# Cardiac Computed Tomography Angiography in CAD Risk Stratification and Revascularization Planning

**DOI:** 10.3390/diagnostics13182902

**Published:** 2023-09-11

**Authors:** Chirag R. Mehta, Aneeqah Naeem, Yash Patel

**Affiliations:** Department of Cardiology, The Warren Alpert Medical School of Brown University, Providence, RI 02903, USAypatel@lifespan.org (Y.P.)

**Keywords:** CCTA, coronary artery disease, risk stratification, revascularization planning

## Abstract

Purpose of Review: Functional stress testing is frequently used to assess for coronary artery disease (CAD) in symptomatic, stable patients with low to intermediate pretest probability. However, patients with highly vulnerable plaque may have preserved luminal patency and, consequently, a falsely negative stress test. Cardiac computed tomography angiography (CCTA) has emerged at the forefront of primary prevention screening and has excellent agency in ruling out obstructive CAD with high negative predictive value while simultaneously characterizing nonobstructive plaque for high-risk features, which invariably alters risk-stratification and pre-procedural decision making. Recent Findings: We review the literature detailing the utility of CCTA in its ability to risk-stratify patients with CAD based on calcium scoring as well as high-risk phenotypic features and to qualify the functional significance of stenotic lesions. Summary: Calcium scores ≥ 100 should prompt consideration of statin and aspirin therapy. Spotty calcifications < 3 mm, increased non-calcified plaque > 4 mm^3^ per mm of the vessel wall, low attenuation < 30 HU soft plaque and necrotic core with a rim of higher attenuation < 130 HU, and a positive remodeling index ratio > 1.1 all confer additive risk for acute plaque rupture when present. Elevations in the perivascular fat attenuation index > −70.1 HU are a strong predictor of all-cause mortality and can further the risk stratification of patients in the setting of a non-to-minimal plaque burden. Lastly, a CT-derived fractional flow reserve (FFR_CT_) < 0.75 or values from 0.76 to 0.80 in conjunction with additional risk factors is suggestive of flow-limiting disease that would benefit from invasive testing. The wealth of information available through CCTA can allow clinicians to risk-stratify patients at elevated risk for an acute ischemic event and engage in advanced revascularization planning.

## 1. Introduction

Functional stress testing has remained the standard of care in evaluating stable chest pain and suspected coronary artery disease (CAD). However, an unremarkable stress test cannot exclude subclinical atherosclerotic plaque at high risk for rupture. Coronary CT angiography (CCTA) has been upgraded to a Class I recommendation by the American Heart Association (AHA) for the evaluation of symptomatic patients with low to intermediate pretest probability for anginal chest pain, due to its ability to exclude obstructive CAD (>70% stenosis with poor collateralization) with high negative predictive value [1]. Myocardial ischemia correlates well with the extent of diameter stenosis from coronary plaques, but the imperfect agreement between both has been well established, and nonobstructive CAD is not insignificant, as it bears prognostic utility. In this setting, qualitative atherosclerotic assessment by CCTA prevails over functional testing through the identification of high-risk phenotypic features (HRPs) that, when present, increase the risk for plaque rupture or erosion [2]. Moreover, the added benefit of calcium scoring in cases of uncertain atherosclerotic cardiovascular disease (ASCVD) risk influences the manner by which cardioprotective medications are prescribed. Advances in computational fluid dynamics have also allowed for non-invasive hemodynamic assessment of suspicious lesions, which was once a historic limitation of CCTA [3]. Lastly, even in the setting of no obvious lesions, underlying inflammatory burden, which still places patients at high risk for subsequent coronary events, can still be ascertained.

As such, the goals of this paper are to comprehensively review the wealth of information provided within the CCTA platform, with a focus on calcium scoring, plaque volumetrics, qualifying HRPs (i.e., spotty calcification, low-attenuation plaque, napkin-ring sign, and positive remodeling), and novel metrics such as pericoronary fat inflammation (Figure 1 and Table 1). We then focus on how CCTA can allow for quantitative interrogation of suspicious lesions to guide advanced revascularization planning, such as with fractional flow reserve and endothelial shear stress, as well as CT perfusion. Lastly, we provide the limitations of CCTA to the reader and explore future directions. 

## 2. Coronary Calcium Scoring

Coronary calcium is a commonly used imaging biomarker for CAD risk stratification. The coronary calcium score (CCS) or Agatston score relies on quantification of calcium deposits in the medial coronary wall on non-contrast CT [4] and is currently categorized as a IIa recommendation by the AHA/ACC for guiding risk reduction therapies in asymptomatic patients ≥ 40 years of age without known CAD, or in patients with a family history of hypercholesterolemia or premature CAD, when 10-year ASCVD risk status is uncertain or if additional information is needed to guide clinician–patient risk discussions [5]. The CCS quantifies large calcium deposits as seen with advanced atherosclerosis and stable plaque phenotypes, though it still serves as a representation of overall plaque burden [6]. Weighted density scores can gauge the 10-year risk of incident ASCVD. In general, a cutoff CCS ≥ 100 or >75th percentile correlates with an ASCVD risk of 7.5 percent per the Multiethnic Study of Atherosclerosis (MESA) and is associated with a higher long-term risk of future MACE when retrospectively applied to the Framingham Heart Study (21.2 versus 4.9 percent, hazard ratio [HR] 5.0, 95% CI 2.1–12.7) [7]. As a result, moderate-intensity statin therapy is recommended when achieving this cutoff. Aspirin use has also been associated with reduced ASCVD events with accepted bleeding risk when the CCS ≥ 100, regardless of 10-year ASCVD risk [8]. The data behind statin and aspirin use with mild CAC scores (i.e., 1 to 99 or <75th percentile) remain limited [9,10,11,12]. Preventive therapies are not recommended with CCS scores of 0, though repeating risk stratification after 5 years is appropriate following counseling on lifestyle changes and modification of cardiovascular risk factors (i.e., dieting, regular exercise, smoking cessation, blood pressure, and hemoglobin A1c control). 

## 3. Spotty Calcifications

Microcalcifications are present in the earliest stages of atherosclerosis and develop from the fusion of calcifying extracellular vesicles as a healing response to areas of intense macrophage inflammation. Microcalcifications give rise to spotty calcifications, which incur mechanical stress on the fibrous cap of the fibroatheroma, leading to its debonding or caveolation and increasing rupture risk [13]. In the ROMICAT II trial, spotty calcification conferred a significant relative risk for acute coronary syndrome (ACS) in patients presenting with chest pain that were risk-stratified and triaged using CCTA (RR 37.2, 95% CI 9.1–152.7) [14]. In the more recent ICONIC study, spotty calcification was associated with a greater risk of ACS at 3.4 years (HR 1.54, 95% CI 1.17–2.04) [11]. Spotty calcifications are broadly defined on CCTA as having a radiodensity > 130 HU and a diameter of <3 mm but with calcium burden length < 1.5 times the vessel diameter and width < ⅔ of the vessel diameter, embedded in non-calcified plaque [15]. Spotty calcifications with diameters < 1 mm are the most worrisome because they are more frequently associated with thin cap fibroatheroma (TCFA) compared to their larger-sized counterparts (31% vs. 9%; *p* < 0.05), as corroborated by intravascular ultrasound with radiofrequency backscatter analysis (IVUS-VH) [16]. Recent cases have been made to incorporate coronary vessel circumference when further qualifying spotty calcifications. One study showed that calcium deposits within an arc of <90° have been found in greater average sums during acute myocardial infarctions compared to unstable or stable angina (1.4 ± 1.3 vs. 1.0 ± 1.1 and 0.5 ± 0.8 lesions, respectively; *p* < 0.0005) [17]. 

## 4. Plaque Characterization, Volumetrics, and Growth

Currently, a standardized lexicon for plaque quantity and characterization is lacking. The simplest organization is the visual differentiation between calcified, mixed, and non-calcified plaques, which can then be further typified using CT attenuation mapping. Unfortunately, attenuation thresholds differ across software and the literature. However, two landmark CCTA studies, PARADIGM and, as previously mentioned, the ICONIC trial, used the following breakdown as supported by IVUS data: calcified plaque as >350 HU and non-calcified plaque ranging from −30 to 350 HU, which can be further subdivided into low attenuation necrotic core −30 to 30 HU (discussed in subsequent sections), fibrofatty 30 to 130 HU, and fibrous 131 to 350 HU [11,18]. 

Several validated scores have been devised to gauge the extent of atherosclerotic disease, including the Segment Involvement Score (SIS), Segment Stenosis Score (SSS), and CT-adapted Leaman score. These scores are easy to perform and do carry prognostic value. In one prospective study, patients with nonobstructive but extensive CAD defined as SIS > 4 had similar cardiac event rates compared to those with obstructive, non-extensive disease (14.5 vs. 13.6 per 1000 patients for cardiovascular death or MI and 26.6 vs. 26.2 per 1000 patients for major adverse cardiac events or MACE; *p* = 0.76 and 0.91, respectively) [19]. Importantly, these scoring systems are semi-quantitative and only provide an estimate of disease burden. A true quantitative appraisal can be measured as component percentages from total plaque, as an absolute area or percentage of the total area using 2D axial cross-sections, and volumetrically on a per-lesion, per-coronary segment, or on a per-vessel territory basis. One common convention is the percentage of overall vessel volume occupied by plaque, presented as percent atheroma volume (PAV) or plaque burden. Total plaque volume, especially non-calcified plaque, confers an increased risk for subsequent cardiac events. In a mixed cohort study by de Knegt et al., patients with ACS and chest pain had greater total plaque volume than positive and negative controls (407 mm^3^ vs. 257 mm^3^ and 148 mm^3^, respectively; *p* < 0.001) and increased proportion of non-calcified plaque elements such as necrotic core (20% vs. 17% and 17%, respectively; *p* < 0.001) [20]. Though cutoffs for high-risk plaque volume remain elusive, the results from a pilot study by Dwivedi et al. suggest a value of 4 mm^3^ of low attenuation plaque per mm of the vessel wall as a potential benchmark for prediction of adverse events, though these results need more validation [21].

An important advantage of quantitative CT is the ability to assess for temporal changes in coronary artery plaque. Assessing the natural history of plaque evolution is helpful, since plaque progression is a known independent risk factor for future ACS, with culprit lesions thought to approximately double in the 3 months prior to an adverse cardiovascular event [22,23]. Interestingly, Lee et al. integrated annual plaque progression rates into a prognostication model and found improved predictive performance for adverse clinical events [24]. Given this evidence, quantitative plaque can prove lucrative as a primary endpoint in clinical trials, as it does not rely on long follow-up periods like mortality metrics. Indeed, medications have been assessed taking advantage of this fact. In the PARADIGM study, statin use was shown to slow plaque progression and increase plaque stability, as suggested by an increase in calcified to non-calcified component ratios over a 14-month span [18]. In the double-blind placebo-controlled EVAPORATE trial, the use of the omega-3 polyunsaturated fatty acid ethyl eicosapentaenoate was shown to reduce total plaque and low attenuation plaque volume by 9% and 17%, respectively, after 9 months of therapy [25]. The drawback of temporal plaque evaluation is the generalizability to everyday use, given the high radiation and contrast exposure with serial CT scans and the need for patient adherence regarding follow-up.

## 5. Low Attenuation Plaque and Napkin-Ring Sign

Low attenuation plaque (LAP), traditionally defined as <30 HU on CCTA, represents a lipid-rich lesion with a thrombogenic necrotic core and has been shown to be a strong predictor for acute coronary events independent of CAC and coronary stenosis per the SCOT-HEART trial (HR 1.60, 95% CI 1.10–2.34). The study also found that patients with an LAP burden higher than 4% were at nearly five times higher risk of suffering from a myocardial infarction (HR 4.65, 95% CI, 2.06–10.5) [26]. 

Thinning of the protective fibrous cap can subsequently expose the lipid core, prompting hemostasis, which increases the risk for acute ischemic events. Optical coherence tomography (OCT) is the ideal imaging modality to assess for fibrous cap thickness due to its high resolution [27], as was used in the COMBINE OCT-FFR trial, where higher rates of cardiac events were demonstrated in patients bearing lesions with TCFA compared to those without (HR 5.12, 95% CI 2.12–12.34) [28]. Similarly, in the CLIMA study, fibrous cap thickness of <75 µm for left anterior descending artery lesions was associated with an increased risk of major adverse cardiac events (MACE) at 1 year (HR 4.7, 95% CI 2.4–9.0) [29]. The CCTA correlate of TCFA is believed to be the napkin-ring sign, which can be visible on thin cross-sections < 0.6 mm and represents an area of low-attenuating luminal narrowing surrounded by a thin (<65–75 μm) hyperattenuating rim < 130 HU. This HRP is thought to represent a lipid-rich necrotic core with intraluminal plaque hemorrhage, microcalcifications, or pre-existing plaque rupture [30]. The napkin-ring sign has been shown to have high specificity and positive predictive value for advanced lesions and future coronary events. In a robust systematic review, its presence was associated with an HR of 5.06 (95% CI, 3.23–7.94) for future MACE and corresponded to the culprit lesion, making the biomarker valuable for targeted lesion guidance and as a correlate for unstable plaque [31]. The napkin-ring sign may also be associated with the concomitant presence of additional HRPs, adding multiplicative risk for future coronary events. For instance, in one piece by Kashiwagi et al., the napkin-ring sign was associated with a higher positive remodeling index and lower overall CT attenuation (1.15 ± 0.12 vs. 1.02 ± 0.12, *p* < 0.01 and 39.9 ± 22.8 HU vs. 72.7 ± 26.6 HU, *p* < 0.01) [32]. 

## 6. Positive Remodeling Index

Positive remodeling refers to the expansion of the external elastic membrane area and associated vasa vasorum proliferation. The positive remodeling index is the ratio of the smallest vessel cross-sectional area of the lesion of interest to the proximal reference luminal area and is defined as ≥1.1 [33,34]. The exact mechanisms linking plaque vulnerability to positive remodeling remain unclear. However, most evidence points to inflammation as the common denominator. To elaborate, the work by Pasterkamp et al. describes an association between plaque inflammation and instability (i.e., increased number of macrophages and metalloproteinases as seen prior to plaque rupture) and positive remodeling in femoral arteries [35]. The theory then surmises that positive remodeling occurs as compensation to accommodate the enlarging subintimal plaque from encroaching into the coronary lumen. Thus, positive remodeling may be used as a temporal biomarker for the early, proliferative plaque that has not had time to stabilize. These pathophysiologic mechanisms have been observed in practice. Data from a study that investigated 1059 patients demonstrated a statistically significant increase in positive remodeling index amongst patients who experienced an ACS within both 12 and 24 months [36]. Positive remodeling was associated with an 11-fold increased risk for ACS in the ROMICAT-II trial [2]. Shortly afterward, the PROSPECT trial demonstrated that a positive remodeling index ratio > 1.1 was associated with an increased plaque burden and increased risk of non culprit lesion major adverse cardiac events in trial patients (HR 2.34, 95% CI 1.00–5.44) [37]. 

## 7. Pericoronary Fat Inflammation

CCTA can detect, typify, and assess the functional significance of plaque. However, inflammatory mechanisms in the vascular wall beget HRP features and flow-limiting lesions, which may be missed. This becomes problematic because a large proportion of ischemic events are not preceded by significant plaque disease, but rather from the erosion and subsequent thrombosis of mild coronary plaque. Indeed, in the PROMISE trial, 54% of adverse events occurred in patients without significant flow-limiting disease, suggesting an underlying inflammatory burden of residual risk [38]. Newer modalities within the CCTA toolset have allowed for detailed qualification of coronary inflammation by imaging perivascular fat. A recent discovery is the bidirectional communication between the vascular wall and perivascular adipose tissue (PVAT), which allows for inflammatory signals from the vessel wall to influence PVAT morphology [39,40,41]. Transcriptomic changes have been observed long before in rat models, with reduced adipocyte gene expression and a lower lipid-to-aqueous tissue ratio, which underlie the noticeable differences in weighted CT attenuation gradients quantifiable through the fat attenuation index (FAI), with increasing values corresponding to greater water density [42]. Evidence for the prognostic utility of FAI was provided by the CRISP-CT trial, which demonstrated an increase in all-cause mortality with an elevated FAI around the RCA (HR 1.49, 95% CI 1.20–1.85), LAD (HR 1. 78, 95% CI 1.42–223), and LCx (HR 1.37, 95% CI 1.10–1.70) arteries independent of the presence of HRPs [43]. FAI values > −70.1 HU around the proximal RCA were strongly predictive of all-cause and cardiac mortality irrespective of HRP features as well. Thus, implementing FAI into the CCTA profile can assist in reclassifying a large portion of patients who do not have HRP features and would otherwise be deemed low-risk to higher-risk subgroups that would benefit from an aggressive, personalized management plan.

## 8. Computational Fluid Dynamics for Hemodynamic Evaluation

The performance of CCTA in patients with obstructive CAD remains suboptimal since the hemodynamic significance of such lesions cannot be ascertained, which may prompt unnecessary invasive coronary angiograms (ICA) for lesions without inducible ischemia. Fractional flow reserve (FFR) has helped address this limitation and has been shown to increase the positive predictive value of CCTA. Traditionally, FFR is obtained by pullback during ICA by measuring the ratio of the mean coronary pressure 1–2 cm away from the stenotic lesion or from the most distal evaluable segment to the aortic pressure under hyperemia [44]. Computational fluid dynamics allows for mathematical modeling of coronary flow without adenosine infusions, which is then overlaid on a three-dimensional coronary model obtained by deep learning algorithms (i.e., contouring and segmentation) from the CCTA input. The HeartFlow Analysis is currently the only U.S. FDA-approved platform for these purposes. FFR with CT imaging (FFR_CT_) has been shown to correlate well with its invasive counterpart, enhancing its role in appropriately triaging patients with chest pain [45].

The greatest utility of FFR_CT_ is in patients with intermediate-risk anatomy, broadly defined as luminal stenosis of 30–69% or ≥70% stenosis in vessels excluding the left main or LAD arteries [46]. Patients with FFR_CT_ values of >0.8 can safely undergo medical management, since deferring revascularization in this cohort is associated with lower MACE and cardiac death after 90 days as well as fewer revascularizations needed at 1 year [47]. FFR_CT_ values < 0.75 should prompt discussions on revascularization, with exceptions being distal vessels or side branches that likely have subtended myocardium not suitable for intervention. FFR_CT_ values between 0.76 and 0.80 represent a gray zone and remain an area of debate, since nearly half of lesions do not have ischemic bearing [48]. In this category, decisions to undergo revascularization should be individualized, taking into account HRPs, lesion location, vessel territory, number of lesions, plaque burden, and patient history. Some experts argue for three months of medical optimization first, if not previously attempted, since the benefits following revascularization in these patients are not convincing [49]. Endothelial shear stress (ESS) is a more novel metric derived from computational fluid dynamics that may assist in further qualifying functional lesion stenosis. Briefly, ESS refers to tangential stress applied on the endothelial surface by flowing blood friction and is dependent on blood viscosity, the gradient of blood velocities at the endothelial wall, and luminal diameter [50]. Normal values of ESS (i.e., with laminar flow) range from 1 to 2.5 Pa. Alterations in ESS influence the progression of atherosclerosis. In vitro and animal studies have demonstrated that chronically low ESS, as seen in the lateral walls of bifurcations, is associated with a proatherogenic phenotype due to loss of flow-oriented alignment of endothelial cells, widening gap junctions, increasing LDL-c accumulation, and cyclical progression of inflammation [51,52]. High ESS, on the other hand, threatens plaque rupture by stimulating endothelial cells to produce plasmin, which destabilizes the fibrous cap [53,54,55,56]. In one observational study by Kalikakis et al., functionally significant lesions had higher ESS compared to non-significant lesions (7 Pa vs. 2.6 Pa; *p* < 0.001), suggesting that adding ESS to stenosis severity can improve discrimination of functionally significant lesions [57]. Additional high-risk morphological changes have also been noted in patients with CAD already on statin therapy and high ESS states, including progression of necrotic core and compensatory expansive remodeling [53]. These findings suggest the possible utility of early anti-hypertensive therapy, though more prospective, randomized control data are needed. 

Patients with high-risk anatomy (i.e., left main disease with ≥50% stenosis, high-grade LAD stenosis of ≥70%, or three-vessel disease) should be referred for invasive intervention. In these situations, especially for multivessel disease, a non-invasive assessment beforehand with FFR_CT_ has been shown to reduce incomplete percutaneous revascularization rates and cost while improving quality of life [58,59]. The RIPCORD trial also showed that compared to CCTA alone, the addition of FFR_CT_ changed the vessel for percutaneous coronary intervention (PCI) in 18% of cases [60]. The inputs derived from FFR_CT_ influence candidacy for coronary artery bypass grafting (CABG) as governed by the SYNTAX score, which qualifies CAD complexity by taking into account several lesion features (i.e., bifurcations, complete total occlusions, and vessel tortuosity with scores greater than 23 suggesting derivable benefit from CABG). The results from the SYNTAX III revolution trial demonstrated that compared to CCTA alone, the addition of FFR_CT_ changed the vessel for revascularization in 12% of cases and reclassified 14% of patients to a lower SYNTAX bracket [61]. The procurement of SYNTAX inputs non-invasively obviates the need for invasive FFR, which has non-trivial risks to the patient (i.e., fluoroscopy time, radiation dose, peri-procedural complications, etc.). Figure 2 provides a basic approach pathway for incorporating FFR_CT_ into revascularization planning.

Patients with low-risk anatomy (i.e., no coronary artery disease or lesions with <30% stenosis) can be treated with optimal medical therapy, whereas patients with high-risk anatomy defined as LMCA with ≥50% stenosis, high-grade LAD stenosis ≥ 70%, or three vessel disease, should be referred for invasive angiography. Patients with intermediate risk anatomy benefit the most from implementation of FFR_CT_. If FFR_CT_. is normal, patients can be deferred to medical therapy whereas if FFR_CT_ is abnormal, invasive angiography is the next best step. Those with values of FFR_CT_ 0.76−0.80 should undergo further risk stratification based on ancillary CCTA data and cardiovascular risk factors. Higher risk patients can be reasonably managed invasively whereas lower risk patients can undergo a trial of 3 months of medical therapy. Recalcitrant symptoms should then prompt an invasive angiogram.

Studies validating FFR_CT_ have focused on single epicardial stenoses, for which the hyperemic pressure gradient represents a true FFR. More commonly, however, CAD presents diffusely along the length of the vessel, and the interaction of hemodynamics between lesions precludes reaching maximum theoretical hyperemia, leading to underestimation of actual post-stenotic pressure drops following the most proximal lesion [62]. Thus, the functional contribution of each stenosis is difficult to ascertain, potentially leading to suboptimal revascularization decisions. One exception can be made with the left main coronary artery (LMCA), since the majority of studies have shown that even with obvious disease, if the FFR > 0.85 in an unobstructed side branch, then the assumption can be made that the LMCA lesion is non-significant [63,64,65,66]. The current standard of care involves manual pullback of a pressure wire at key points along the vessel to discern lesions with the largest translesional pressure gradients. Recently, a novel FFR_CT_-based PCI planner tool has been developed that simulates post-revascularization pressure differentials, allowing for accurate prediction of FFR contribution for each stenosis, correlating well with invasive, manual pullback FFR [62]. 

## 9. Myocardial Stress CT Perfusion

CT perfusion (CTP) is another technique in addition to FFR_CT_ for the functional differentiation between nonobstructive and obstructive CAD. During CTP imaging, radiopaque iodinated contrast is measured as it perfuses through the coronary circulation and myocardium. Static or dynamic CTP can be used to assess for myocardial ischemia using vasodilators (e.g., adenosine or regadenoson) at rest and has been shown to be non-inferior to SPECT for detecting reversible ischemia with an agreement rate of 0.87 [67]. Reversible myocardial perfusion defects during stress are identified as areas of low attenuation, whereas fixed perfusion defects corresponding to scar tissue from previous infarcts also present as areas of decreased enhancement but have delayed washout, as seen on dynamic CTP. Along with FFR_CT_, CTP is complementary by conferring greater specificity and positive predictive value for detecting flow-limiting CAD, especially when the anatomical imaging is inconclusive, such as with moderate coronary stenosis (50–70%). Indeed, in the multicenter trial CORE320, stress CT perfusion and CCTA combined had greater accuracy in discriminating for flow-limiting disease compared to CCTA alone [68]. Moreover, in an earlier paper by Ko et al., the combination of ≥50% stenosis on CTCA and a perfusion defect on stress CT perfusion was 98% specific for an invasive FFR value ≤ 0.8, whereas <50% stenosis on CCTA and normal stress CTP were 100% specific for non-ischemic disease [69].

There is limited head-to-head data on CT perfusion vs. FFR_CT_ for functional assessment of suspicious lesions, and the clinical applications of each modality will depend on their strengths and weaknesses. For instance, stress CTP may be appropriate in cases with significant coronary calcification due to blooming artifact, given that FFR_CT_ uses CCTA images as boundary conditions for computational fluid dynamics analysis. CTP is also more appropriate in situations with inadequate patient cooperation due to motion artifacts and the need for breath-holding in FFR_CT_. On the other hand, FFR_CT_ may be more beneficial in cases of 3-vessel disease, since CTP may not unmask balanced ischemia [70]. Data are similarly limited on the concomitant use of CTP and FFR_CT_ for functional assessment. However, a study by Coenen et al. suggested that a combination of both modalities may be superior to either technique alone [71]. More prospective randomized controlled data are needed on this end. 

## 10. Limitations of Ccta

Limitations of performing CCTA include the long turnaround time, which is not practical in the emergency setting. Images can be hindered by motion artifacts. Temporal resolution can be affected by suboptimal heart rates > 60 bpm and respirophasic variations. The presence of severe calcification can lead to blooming artifacts and obscure vessel boundaries (i.e., beam hardening and related halo from high-attenuating structures). Care must be taken in patients with acute or chronic kidney injury due to the risk of contrast-induced nephropathy. Lastly, radiation exposure is always a concern, though high-pitch acquisition platforms have allowed for end-diastolic and end-systolic capture in a single heartbeat with sub-millisievert exposure [72].

## 11. Future Directions

Machine learning (ML) algorithms, particularly radiomics, are currently being developed to assist in the seamless identification of vulnerable plaque. Radiomics refers to the collection of automated high-throughput techniques that extract geographic properties and textural characteristics from medical imaging [73]. Recently, radiomics were used for feature extraction of pericoronary fat from adipose tissue biopsies. The radiomic fat phenotype was then used as an input to train supervised ML algorithms to identify adipose tissue inflammation, which was externally validated on the cohorts from the CRISP-CT and SCOT-HEART studies [74]. As such, there is tremendous applicability of ML in elucidating more complex alterations at the morphological level, unveiling hidden, novel biomarkers. Rigorous validation will be needed, however, with randomized prospective studies.

## 12. Conclusions

Flow-limiting coronary disease through functional stress testing serves well to target patients who would benefit from revascularization. However, stress testing cannot qualify nonobstructive coronary disease, which is not insignificant. The armament of features within the CCTA toolset allows for imaging analysis beyond the dichotomy of obstructive and nonobstructive. Having the knowledge of coronary calcium burden and high-risk phenotypic features invariably affects primary prevention strategies. The addition of FFR_CT_ and CTP offers guidance for revascularization planning and improves the specificity of CCTA for detecting lesion-specific ischemia, enhancing its role as a gatekeeper for ICA. CCTA can thus provide a comprehensive, non-invasive assessment for optimal, individualized coronary care.

## Figures and Tables

**Figure 1 diagnostics-13-02902-f001:**
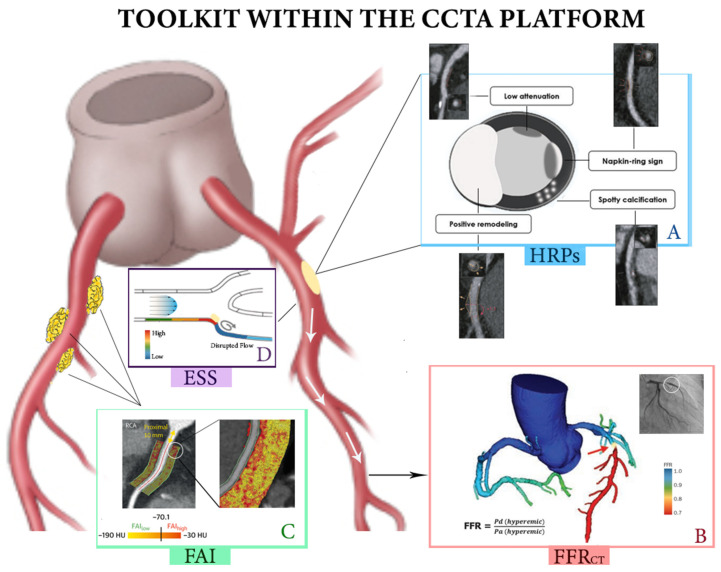
Summary diagram of the CCTA toolset. (**A**) CCTA can be used to characterize vulnerable nonobstructive plaque. The four cardinal HRPs, as demonstrated through a proximal LAD lesion, include spotty calcifications, positive remodeling, low attenuation, and the napkin-ring sign. (**B**) FFR_CT_ can be used to gauge the hemodynamic significance of obstructive lesions, with the greatest utility in patients with intermediate risk anatomy. In this example, the proximal LAD lesion has a pressure drop with associated FFR_CT_ < 0.7, suggesting derived benefit from invasive angiography. (**C**) FAI assists in reclassifying patients without significant flow-limiting disease by qualifying coronary inflammation via attenuation gradients in PVAT. In this example, certain regions of fat deposits around the proximal RCA have regions of high inflammatory burden, quantified as >−70.1 HU. (**D**) ESS is another utility of CCTA. ESS is defined as a pressure (Pa or dynes/cm^2^). High ESS before a lesion is associated with plaque rupture and subsequent thrombosis. Low ESS downstream of a lesion secondary to eddy currents is associated with plaque progression.

**Figure 2 diagnostics-13-02902-f002:**
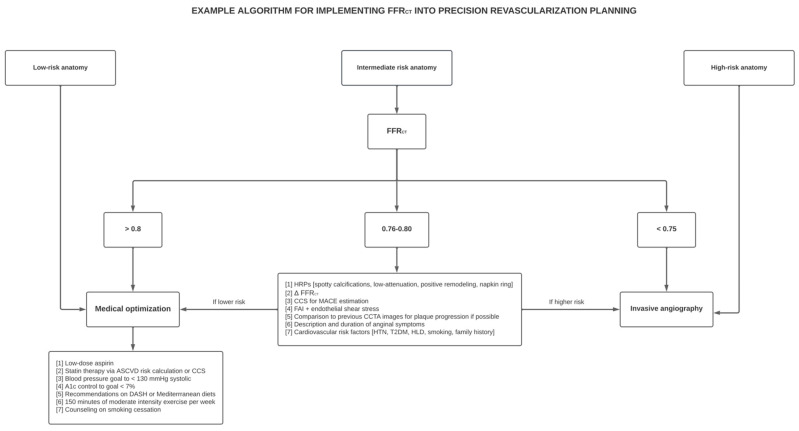
Basic approach flowchart towards integration of FFR_CT_ in revascularization planning for patients with atypical chest pain.

**Table 1 diagnostics-13-02902-t001:** Summative organization of HRPs and additional metrics extractable from CCTA to further aid in CAD risk stratification and revascularization planning.

HRP	Key CCTA Findings	Significance
Spotty calcifications	▪ Calcified lesion > 130 HU ▪ Diameter of <3 mm but <1 mm most worrisome ▪ Arc < 90° ▪ Embedded in non-calcified plaque	Spotty calcifications incur mechanical stress on the fibrous cap, increasing the risk for plaque ruptures as highlighted in the ROMICAT II and ICONIC trials
Positive remodeling index	The positive remodeling index is quantitatively defined as the ratio of the smallest lesion cross sectional area to proximal reference luminal area, with a value ≥ 1.1 considered as elevated.	Positive remodeling refers to expansion of the external elastic membrane area, with intimal neovascularization and vasa vasorum proliferation as compensation to preserve luminal patency from encroaching subintimal plaque and has been associated with an 11x risk of ACS per the ROMICAT II study.
Low attenuation plaque (LAP)	Defined as < 30 HU	LAP represents a lipid rich lesion with thrombogenic necrotic core and is a strong predictor for future ACS per the SCOT-HEART trial.
Napkin-ring sign	Luminal narrowing caused by a low attenuation lesion that is surrounded by a thin (65–75 μm) hyperattenuating rim < 130 HU	The napkin ring sign is thought to represent intraluminal plaque hemorrhage, microcalcifications, or pre-existing plaque rupture, and is the closest correlate of the TCFA on OCT, which has demonstrated association with MACE per COMBINE OCT-FFR and CLIMA trials.
Additional Metrics Not Defined as HRPs Within the CCTA Toolset		
Plaque volumetrics	One pilot study suggested 4 mm^3^ of low attenuation plaque per mm of vessel wall as a potential benchmark for prediction of adverse events	Patients with ACS had statistically significant elevations in total plaque volume and non-calcified plaque compared to positive and negative controls in a post-hoc analysis of the CATCH and VERDICT trials by de Knegt et al.
Pericoronary fat inflammation	Inflammation within the pericoronary fat can be assessed by differences in attenuation and quantified through the FAI with a value > −70.1 HU as a high-risk benchmark.	Attenuation within PVAT can be used as a surrogate for residual inflammatory burden and assist in reclassifying lower risk patients.
Computational fluid dynamics	Computational fluid dynamics, specifically fractional flow reserve and ESS, allow for hemodynamic assessment of lesions concerning for inducible ischemia. CT-derived fractional flow reserve < 0.75 or values from 0.76–0.80 in conjunction with additional risk factors such as high ESS, is suggestive of flow-limiting disease that would benefit from invasive testing.	The use of computational fluid dynamics can transform CCTA as a [1] gatekeeper for ICA and [2] a roadmap of the coronary tree to assist in advanced revascularization planning

## Data Availability

No new data were created in this study. Data sharing is not applicable to this article.

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
