# Peer review of "Cardiac Computed Tomography Angiography in CAD Risk Stratification and Revascularization Planning"

_diagnostics, 2023, doi:10.3390/diagnostics13182902_

Round 1
Reviewer 1 Report
Thank you for the opportunity to review your manuscript entitled "Cardiac Computed Tomography Angiography in CAD Risk 2 Stratification and Revascularization Planning ".Abstract, title and references.
The aim of the study is clear. The title is informative and relevant. The references are relevant, recent, and referenced correctly.
Introduction.
It is clear what is already known about this topic. The research question is clearly outlined.
Methods and Results.
The process of subject selection is clear. The variables are defined and measured appropriately. The study methods are valid and reliable. There is enough detail in order to replicate the study.
Discussion.
The manuscript is well written and a stimulus for the readership. It is worth asking, however, whether the level of cardiac markers such as Troponin T and NtproBNP was assessed?
If so, were there differences in levels between the groups?
Thank you
Author Response
Thank you for the feedback! From what we understand, the Troponin T and NtproBNP were not assessed between the groups.
Reviewer 2 Report
Dear Authors,
I would like to thank you for submitting the manuscript entitled: "Cardiac Computed Tomography Angiography in CAD Risk Stratitifaction and Revascularization Planning" to Diagnostics Journal.
There are extremely rare situations while reviewing the manuscript I have nothing to add except of congratulations to the authors.
Kinds
T
Author Response
Thank you for the feedback!
Reviewer 3 Report
I found this to be a well organized and well written review. The subject is reviewed in a comprehensive manner studying the present status and possible future of CCTA. I believe the positives and pitfalls were honestly presneted.
Author Response
Thank you for the feedback!
Reviewer 4 Report
The work is well structured with a sufficient characterization of the topics dealt in the different chapters; it, actually, wants to represent an interesting contribution in representing high risk phenotypic features (HRP) and key coronary CT angiography (CCTA) findings.
It is important to remember that in the clinical field of application, since 2019, European Society of Cardiology guidelines for the diagnosis and management of chronic coronary syndromes classified CCTA as a Class 1 recommendation for diagnosing coronary artery disease (CAD) in symptomatic patients with suspected obstructive coronary artery disease.
The guideline further states that CCTA should be preferentially considered if there is a low likelihood of obstructive CAD, if the patient characteristics suggest high image quality, if a local expertise is available, and finally if an appropriate level of information on atherosclerosis is desired, and there is no history of CAD. These issues should represent the main criteria in the assesment of the criteria of these diagnostic techniques.
The Authors opportunely underlined as CCTA has undergone remarkable technological advances, as you can see in the wide analysis of HRP Moreover the technique has led to improved image quality suitable for the application of new investigational analysis. But the most important reason is its ability to extract functional information from routine CCTA and further elaborations in prognostic and therapeutic fields; in my opinion the Authors should better explain the clinical applicability of a lot of the examined HRP.
In the clinical cardiological setting, it is known that CCTA has evolved into a truly versatile imaging modality that can depict atherosclerosis burden, determine functional significance of a stenotic lesion, and guide the management and treatment of stable coronary artery disease; otherwise they don’t have full consciousness yet of the new reported techniques generally not reported in the diagnostic reports.
Several other studies with large-scale CCTA registries have previously shown the cardiovascular risk of non obstructive CAD. Given that statin and aspirin prescriptions significantly increase upon abnormal findings by CCTA like the presence of atherosclerosis, tailored treatment for non obstructive CAD would improve the prognosis of these patients and contribute to a better patient management, after the achievement of an appropriate risk stratification.
Additionally, the Author don’t evaluate in appropriate way, in my opinion, the role of Cardiac CT perfusion imaging, that allows comprehensive noninvasive functional assessment of CAD, similar to nuclear perfusion imaging with SPECT .
It’s known that myocardial CT perfusion imaging has several advantages over FFRCT in that it is not affected by calcified plaque, patient motion, or inadequate patient cooperation during breath holding. Adding CT perfusion imaging to standard CCTA also improves diagnostic accuracy for identifying hemodynamically significant obstructive coronary artery stenosis.
Finally, some considerations could be reserved to technological devices availability; we know that, compared to the prior 64-slice multidetector CT, newer-generation scanners include different features such as improved spatial and temporal resolution, faster scan mode, and whole heart coverage with either wide-detector or dual-source CT; these devices are able to acquire images of the entire heart in a single beat and to obtain ultra-high-pitch mode scan mode, that can image the heart in less than 300 ms.. Furthermore, thinner detectors with a spatial resolution of 250 microns along the XY planes, faster gantry rotation (220 ms), noise reduction with improved detector efficiency, and innovative electronic circuitry have made it possible to obtain diagnostic images in patients who were once deemed too challenging to image with prior-generation CT (eg, those with calcium score > 400 AU, large coronary artery stents, coronary artery bypass grafts, heart rate > 65 bpm, atrial fibrillation, and obesity/body mass index > 30 kg m−2). So, in my opinion, it is important that the characteristics of the devices used to acquire the various HRPs on display be made explicit.
Certainly, for the clinician it is not sufficiently clear and deducible how much the evaluation methods expressed represent on the one hand acquired and consolidated criteria, other criteria in the application phase.
I don't know if the authors represent a working group of radiologists or if they also include clinical cardiologists/imaging experts, but certainly there is an evident gap between what is widely described and represented and what is usually the subject.
The work is well structured with a sufficient characterization of the topics dealt in the different chapters; it, actually, wants to represent an interesting contribution in representing high risk phenotypic features (HRP) and key coronary CT angiography (CCTA) findings.
It is important to remember that in the clinical field of application, since 2019, European Society of Cardiology guidelines for the diagnosis and management of chronic coronary syndromes classified CCTA as a Class 1 recommendation for diagnosing coronary artery disease (CAD) in symptomatic patients with suspected obstructive coronary artery disease.
The guideline further states that CCTA should be preferentially considered if there is a low likelihood of obstructive CAD, if the patient characteristics suggest high image quality, if a local expertise is available, and finally if an appropriate level of information on atherosclerosis is desired, and there is no history of CAD. These issues should represent the main criteria in the assesment of the criteria of these diagnostic techniques.
The Authors opportunely underlined as CCTA has undergone remarkable technological advances, as you can see in the wide analysis of HRP Moreover the technique has led to improved image quality suitable for the application of new investigational analysis. But the most important reason is its ability to extract functional information from routine CCTA and further elaborations in prognostic and therapeutic fields; in my opinion the Authors should better explain the clinical applicability of a lot of the examined HRP.
In the clinical cardiological setting, it is known that CCTA has evolved into a truly versatile imaging modality that can depict atherosclerosis burden, determine functional significance of a stenotic lesion, and guide the management and treatment of stable coronary artery disease; otherwise they don’t have full consciousness yet of the new reported techniques generally not reported in the diagnostic reports.
Several other studies with large-scale CCTA registries have previously shown the cardiovascular risk of non obstructive CAD. Given that statin and aspirin prescriptions significantly increase upon abnormal findings by CCTA like the presence of atherosclerosis, tailored treatment for non obstructive CAD would improve the prognosis of these patients and contribute to a better patient management, after the achievement of an appropriate risk stratification.
Additionally, the Author don’t evaluate in appropriate way, in my opinion, the role of Cardiac CT perfusion imaging, that allows comprehensive noninvasive functional assessment of CAD, similar to nuclear perfusion imaging with SPECT .
It’s known that myocardial CT perfusion imaging has several advantages over FFRCT in that it is not affected by calcified plaque, patient motion, or inadequate patient cooperation during breath holding. Adding CT perfusion imaging to standard CCTA also improves diagnostic accuracy for identifying hemodynamically significant obstructive coronary artery stenosis.
Finally, some considerations could be reserved to technological devices availability; we know that, compared to the prior 64-slice multidetector CT, newer-generation scanners include different features such as improved spatial and temporal resolution, faster scan mode, and whole heart coverage with either wide-detector or dual-source CT; these devices are able to acquire images of the entire heart in a single beat and to obtain ultra-high-pitch mode scan mode, that can image the heart in less than 300 ms.. Furthermore, thinner detectors with a spatial resolution of 250 microns along the XY planes, faster gantry rotation (220 ms), noise reduction with improved detector efficiency, and innovative electronic circuitry have made it possible to obtain diagnostic images in patients who were once deemed too challenging to image with prior-generation CT (eg, those with calcium score > 400 AU, large coronary artery stents, coronary artery bypass grafts, heart rate > 65 bpm, atrial fibrillation, and obesity/body mass index > 30 kg m−2). So, in my opinion, it is important that the characteristics of the devices used to acquire the various HRPs on display be made explicit.
Certainly, for the clinician it is not sufficiently clear and deducible how much the evaluation methods expressed represent on the one hand acquired and consolidated criteria, other criteria in the application phase.
I don't know if the authors represent a working group of radiologists or if they also include clinical cardiologists/imaging experts, but certainly there is an evident gap between what is widely described and represented and what is usually the subject.
Author Response
A paragraph on CT myocardial perfusion has been added. The comments regarding the technological device innovations are certainly novel. However, given what was previously ascribed to us when invited to write the review [CCTA use for risk stratification and planning], our team feels strongly that technological innovations to CT is likely outside the scope of this topic review and based on how much data there is currently, this topic can likely be a review paper on its own.
Round 2
Reviewer 3 Report
Tou have addressed my concerns
Reviewer 4 Report
The current review presents a very appropriate integration for clinical relapses in particular.
Even the changes in the setting of the chapters and those implemented in the introduction are in line with an improvement especially in terms of understanding.
In my opinion, the current structure of the text is more complete and in line with the purpose of the work and what is represented in the title.
The current review presents a very appropriate integration for clinical relapses in particular.
Even the changes in the setting of the chapters and those implemented in the introduction are in line with an improvement especially in terms of understanding.
In my opinion, the current structure of the text is more complete and in line with the purpose of the work and what is represented in the title.